# Social dilemmas and poor water quality in household water systems

Gopal Penny[1,2,3], Diogo Bolster[1,2], and Marc F Müller[1,2]

[1]Civil and Environmental Engineering and Earth Sciences, University of Notre Dame, Notre Dame, Indiana, USA
[2]Environmental Change Initiative, University of Notre Dame, Notre Dame, Indiana, USA
[3]Department of Geography, National University of Singapore, Singapore, Singapore

**Correspondence:** Gopal Penny (gopalpenny@gmail.com)

**Abstract.** Private water supply systems consisting of a domestic well and septic system are used throughout the world where households lack access to public water supply and sewers. In residential areas with high housing density, septic contamination of private wells is common and associated with multiple health concerns. This situation can give rise to social dilemmas, where individual costs dis-incentivize homeowners from investing in enhanced septic systems that would reduce well contamination and bring communal benefits. We combine a stylized game-theoretical model with a probabilistic groundwater model to characterize how economic and hydrogeological conditions interact to produce misaligned incentives conducive to social dilemmas. The occurrence of social dilemmas depends on the relative costs of well contamination versus the cost of installing an enhanced septic treatment system, and the relative probabilities of cross-contamination versus self-contamination. The game reveals three types of social dilemmas that occur in such systems, with each calling for distinct policy solutions. We demonstrate how the model can be applied to existing systems using a case study of St Joseph County, Indiana, where high nitrate contamination rates have raised public health concerns. This analysis represents a step towards identifying alternative policy solutions for a problem that has remained difficult to address for decades.

## 1 Introduction

Groundwater plays a critical role in supporting social and ecological systems throughout the world (Gleeson and Richter, 2018). Groundwater provides over 50% of urban water supply (Zektser and Everett, 2004) and 40% of water for irrigation (Siebert et al., 2010). Protecting the groundwater commons has become increasingly important and challenging as aquifers deplete and become contaminated (Gleeson et al., 2020; Hartmann et al., 2021). Indeed, contamination has become prevalent in groundwater systems throughout the world (Charalambous, 2020). In many cases, regulatory frameworks struggle to provide effective preventative measures and the challenge of groundwater protection transforms into a problem of groundwater remediation (Nieto et al., 2005; Hou et al., 2018). Fundamental to the challenge of groundwater protection is the issue of environmental externalities, where the polluter gains some benefit from inadequate (or faulty) treatment prior to water being discharged or leaked to the subsurface (Hellegers et al., 2001). Although this problem arises in a variety of industrial and agricultural settings, groundwater systems that supply residential communities can also be compromised by inadequate household water systems (Withers et al., 2014).

Privately supplied household water systems consisting of a domestic well and septic system are common throughout the world in areas without piped supply and sewers. In the United States, approximately 13% of the population relies on water supply from a private domestic well (Dieter et al., 2018, also see Fig. 1) and 25% of households discharge wastewater through a septic system (US EPA, 2005a). Although the installation of septic systems is generally governed by a combination of guidelines from state and local governments (Thomassey and Dutcher, 2017), over half of septic systems were installed at least 30 years ago and up to 20% are malfunctioning with potentially more underperforming (US EPA, 2005a). Domestic wells are unregulated and the responsibility for maintaining water quality is left to each individual household (Bowen et al., 2019). Privately supplied household water systems are at a relatively high risk of contamination from a variety of sources (Bastani and Harter, 2019), and water quality in private systems often fails to match water quality in highly-regulated public water supply systems (Focazio et al., 2006; DeSimone et al., 2009). These risks are compounded in low-income households, which often face higher risk in water quality than higher-income households, disproportionately burdening segments of the population (Schaider et al., 2019; Meehan et al., 2020a). Overall, surveys indicate that 2-14% of private wells are contaminated with nitrates (Hoppe et al., 2011), but this can rise considerably higher in locations with high housing density (Yates, 1985; Bremer and Harter, 2012).

Septic tanks represent one of the major threats to water quality in domestic wells and the high concentration of contaminants in septic leachate creates a variety health concerns (US EPA, 2005b; Katz et al., 2011). Septic contamination has been associated with endemic diarrheal illness in Wisconsin (Borchardt et al., 2003), viral discharge and transport exceeding typical well-drainfield setback distances (DeBorde et al., 1998), and multiple disease outbreaks in the United States and Canada (Craun

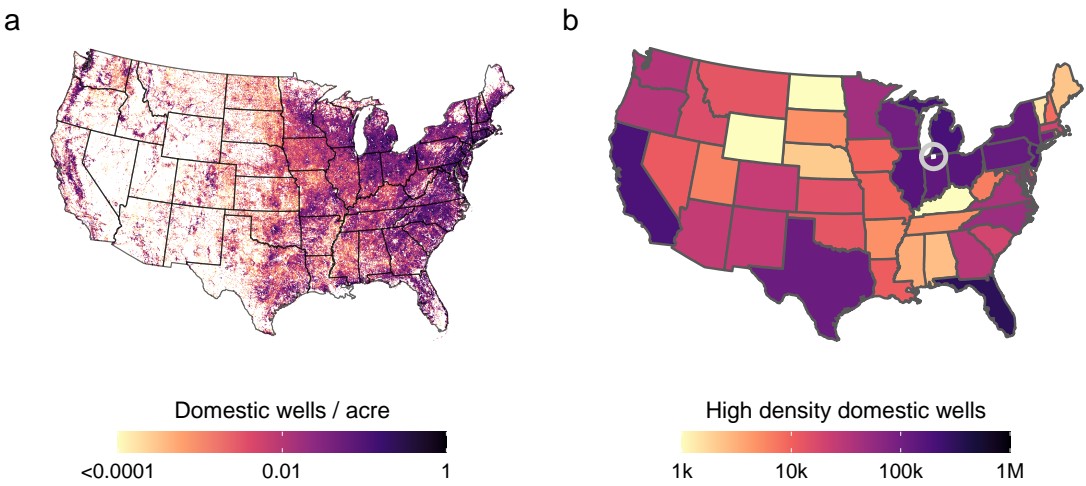

**Figure 1.** Concentration of households with domestic wells in the contiguous United States. (a) Gridded (1 km) density of wells. (b) Total number of high density wells by state, defined as wells (in 1 km pixels) where the average density is at least 1 per acre. Data are from Johnson et al. (2019). The location of the case study (Sect. 4) is marked inside the gray circle.

et al., 1994; Beller et al., 1997). In addition to pathogens, septic systems also discharge high levels of nitrates, which can cause acute health problems in infants (Knobeloch et al., 2000), and other emerging contaminants such as PFAS, flame retardants, and endocrine disruptors (Schaider et al., 2016), which also present long-term exposure risks (Bergman et al., 2013).

The issue of septic contamination has been widely recognized for at least half a century (Yates, 1985). While technological solutions (e.g., reverse osmosis) are available to treat water from contaminated wells, due to high costs such solutions are often limited to wealthier households and only implemented at select points of consumption (e.g., kitchen sinks), leaving potential contamination pathways open through other access points. Furthermore, a well contaminated with nitrate pollution can reduce the value of a home by up to 6% (Guignet et al., 2016). Additional options exist to reduce or remove contaminants at the source. For instance, traditional septic systems are designed to mitigate the effect of pathogens, and advanced septic systems can remove nitrates from septic leachate (Washington State Department of Health, 2005). However, the uncertainty of groundwater flow, combined with the lay knowledge of most homeowners, means that residents often have little clarity about the source of contamination of their own well. Given that contamination could originate from a neighboring well, homeowners may lack incentives to maintain or upgrade their septic system even if doing so would decrease pollution risk for all residents. This discrepancy means that the best choice for each individual household may be in conflict with the best choice for the community as a whole. Such scenarios are known as social dilemmas because the solution that maximizes the welfare of the community does not correspond to outcomes that arise from individual community members acting rationally and independently (Kollock, 1998).

The management challenge pertaining to household water systems has focused primarily on understanding the water quality, public health, and technical aspects of these systems including source identification (Zendehbad et al., 2019) and treatment options (Juntakut et al., 2020). These analyses are directly related to questions of the severity, risk (Li et al., 2019), and the extent to which such systems should be regulated (Bremer and Harter, 2012). Here, we re-frame the problem from the perspective of household utility, with the goal of identifying when households may benefit from policy instruments to support solutions that maximize collective welfare. To address this challenges, we develop a game theoretic model to understand how social dilemmas might arise in household water systems. Framing pollution and treatment behavior through game theory has been done in a variety of case studies (Madani, 2010; Dinar and Hogarth, 2015). However, most game theoretic models of water quality focus predominantly on surface water quality (Šauer et al., 2003; Schreider et al., 2007; Estalaki et al., 2015) including, for instance, the concentration of pollutants in groundwater recharge (Raquel et al., 2007). We are unaware of any studies that couple game theory and groundwater modeling to assess water quality in household water systems with a domestic well and septic field. In order to frame the problem in quantitative terms, we consider the net utility, including the costs and benefits of water protection (as identified by Raucher, 1983; Crocker et al., 1991), and the perceived economic value of clean groundwater (Caudill and Hoehn, 1992; Brouwer and Neverre, 2020; Charalambous, 2020).

We proceed to develop theory regarding social dilemmas in communities with predominantly private water systems. We begin by focusing on two-household scenarios to capture different social dilemmas that could occur in such simple configurations (Sect. 2) and how player preferences depend on the cost of septic treatment as well as the cost of contaminated well water. We then build on this framework to understand how social dilemmas arise in N-player games with varying housing density

(Sect. 3). In order to evaluate the potential for social dilemmas in a real-world scenario, we apply the model to a case study in St. Joseph County, Indiana (Sect. 4) where nitrate contamination in privately supplied household water systems has been an ongoing concern for the county health department. Finally, we review the different types of social dilemmas that may occur in such household water systems, along with a discussion of barriers and policy opportunities to support effective management strategies (Sect. 5).

## 2 Two-household contamination games

### 2.1 Expected utility and payout matrices

We first consider a two-player static game with complete information, wherein the payout structure of both players is common knowledge. Note that we employ non-cooperative game theory to determine the presence or absence of social dilemmas, rather than the manner in which players build coalitions (as in cooperative game theory). As described below, this approach is realized by comparing outcomes under individual optimization (the Nash equilibrium) and full cooperation (the social optimum). Because the payout structure for all players is common knowledge, it is a game of complete information. In the game, each player chooses whether to upgrade their septic system to an enhanced system with contaminant removal (E) at some cost, or to keep a basic septic system (B) at no cost. The cost of upgrading, $C_\sigma$, represents the difference between the enhanced septic system and the cost for a conventional system (e.g., as mandated by local regulations). If a player's well becomes contaminated, that player also incurs a cost, $C_x$. Although the game can be played at any time, it is a static one-shot game because the action to upgrade a septic system only occurs once via capital investment and multiple stages (as in a dynamic game) are not necessary to determine the possibility of a social dilemma. The expected utility of an individual household $i$ is therefore

$$\mathbb{E}[U_i] = -\sigma_i C_\sigma - p_i C_x \,, \tag{1}$$

where $\sigma_i \in \{0, 1\}$ represents the action space of each household, which can choose to maintain a basic septic system (B, $\sigma_i = 0$) or upgrade to an enhanced system (E, $\sigma_i = 1$). The probability $p_i$ describes the likelihood of contamination of player i's well, which depends on the septic system decisions made by *both* players ($\sigma_i$ and $\sigma_j$) and on the groundwater flow direction. Lastly, the cost of contamination, $C_x$, can either represent the individual cost of household water treatment to improve household water quality (which represents a lower bound on the cost, Yadav and Wall, 1998), or the cost of using contaminated water and associated consequences (Raucher, 1983). Note that no player would benefit from upgrading their system if doing so was more expensive the cost of contamination. To avoid this trivial outcome, we consider only situations with $C_x$ larger than $C_\sigma$. We evaluate this requirement and explore associated costs of nitrate contamination in greater detail in Sect. 4.

Because Eq. (1) depends on binary decisions taken by two interacting players, its outcomes can be represented as a $2 \times 2$ matrix. The rows and columns of the matrix represent the decision by players 1 and 2, respectively, on whether or not to upgrade. Each cell of the matrix is populated by a pair of normalized payouts representing the expected utility of each player

for each combination of decisions. Payouts from Eq. (1) are normalized so that the summed total payout across players for each cell ranges between 0 and 2. Two types of matrix cells are particularly noteworthy:

- The Nash Equilibrium (NE) represents each player's best (i.e., utility maximizing) response to the other player's own utility-maximizing decision.

- The Social Optimum (SO) represent the combination of decision that maximizes the summed expected utility of both players.

Whether or not the NE and SO decision outcomes overlap depends on the ratio of costs (i.e., the "cost ratio", $C_x/C_\sigma$) and the groundwater configuration between households. We define a social dilemma as a situation where the SO outcome is different from at least one NE outcome. We proceed to enumerate the different configurations of NE and SO in the $2 \times 2$ games that emerge from Eq. (1) for symmetric cases (Sect. 2.2), and in the situation where the two players face different and uncertain costs (Sect. 2.3). In both situations, we identify and characterize emerging social dilemmas.

## 2.2 Symmetric two-player games

If both players face identical costs ($C_x$ and $C_\sigma$) and contamination by (and of) each player is symmetric, four different game configurations emerge depending on the spatial configuration of the two players (Fig. 2). If neither player contaminates their own well or that of the other player, neither player wishes to upgrade to an enhanced septic system and the Nash Equilibrium is identical to the Social Optimum (Fig. 2a). When both players contaminate only their own wells, they are both incentivized to upgrade their septic system and the Nash Equilibrium and Social Optimum solutions are again identical (Fig. 2b).

A social dilemma occurs when individual optimization leads to a different outcome than would be achieved if players work together. This can only occur when players potentially contaminate each other (as in Fig. 2c and 2d). When both players only contaminate the other player's well without contaminating their own well (Fig. 2c), the best response for each player would be not to upgrade (B), regardless of the decision of the other player. While neither player directly gains by upgrading their own septic system, each player would benefit if the other player upgrades. This leads to a classic Prisoner's dilemma (see Kollock, 1998) where neither player upgrades in the Nash equilibrium despite the mutual benefits of doing so. In the final situation (Fig. 2d), players contaminate both their own and the other players' well. This game yields two Nash Equilibria, including one where neither player upgrades (B, B) and another where both players upgrade (E, E). This structure follows an Assurance or Stag-Hunt game (Kollock, 1998), where players only wish to upgrade to an enhanced system when the other player also upgrades. The game has two Nash Equilibria, where either both players hunt the stag or both players hunt the hare. The first Nash Equilibrium (B,B) is not a Social Optimum and therefore creates a social dilemma.

## 2.3 Uncertainty and asymmetry in groundwater behavior

In addition to these symmetric games with complete information, households may have uncertainty with respect to groundwater behavior. We consider five different configurations of households (Fig. 3, I-V) and allow groundwater to flow west, east, or north, with corresponding probability $P(West)$, $P(East)$, and $P(North)$. Allowing flow towards the south would produce

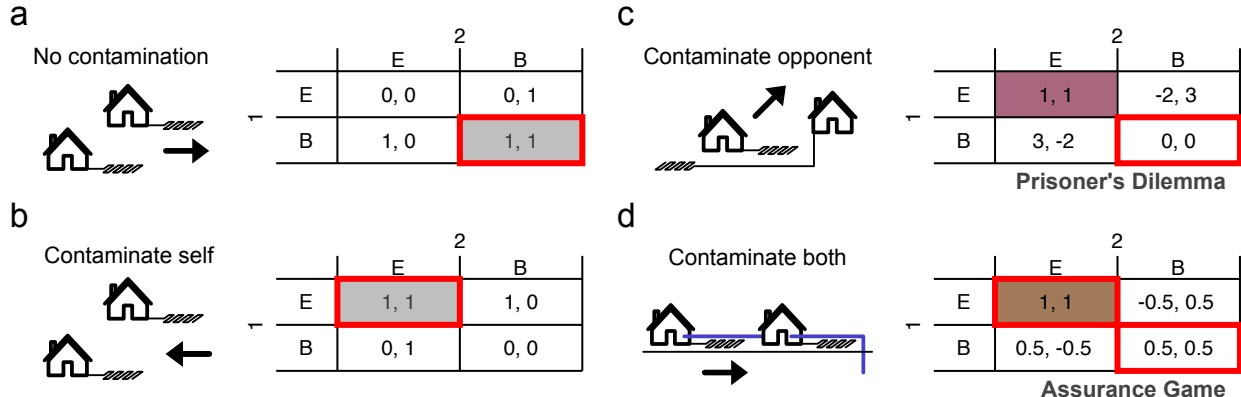

**Figure 2.** Payouts and social dilemmas in symmetric games. In each scenario, the choice to keep a basic septic system results in (a) no contamination, (b) self-contamination, (c) contamination of the other player, and (d) contamination of both players. Payouts represent the situation where $C_x = 1.5C_\sigma$. Utility is normalized so that the social optimal scenario produces an average payout of 1, and the joint worst scenario produces an average payout of 0. Arrows indicate direction of groundwater flow. For scenarios (a) and (b) the Nash equilibrium (red outline) and social optimal (gray shading) are equivalent. A social dilemma arises for scenarios (c) and (d), as there exists a difference between (at least one) Nash equilibrium and the social optimum (red for Prisoner's dilemma and orange for Stag-Hunt).

identical game structures as flow towards the north, and we exclude this unnecessary possibility. The expected utility can be calculated as the weighted average utility from the end-member scenarios. For instance, consider scenario II in Fig. 3. If $P(West) = 0.4$, $P(East) = 0$, $P(North) = 0.6$, and neither player upgrades their septic system, the expected utility for each player would be $0.4C_x$.

Considering all combinations of household configurations and flow probabilities (i.e., with $P(North), P(West), P(East) \in \{0, 0.2, 0.4, 0.6, 0.8, 1\}$), this approach generates three types of social dilemmas (Fig. 3, right): the aforementioned Prisoner's Dilemma (PD) and Stag-Hunt game (SH), and an additional situation that we refer to as an Asymmetric Dilemma (AD). In the latter dilemma, the social optimum differs from the Nash equilibrium which exhibits asymmetric payouts favoring one player over the other.

To better understand how asymmetric dilemmas arise, consider configuration I as an example, where $P(West) = 0.2$, $P(East) = 0.8$, and $P(North) = 0$ (i.e., marked by the $+$, $\circ$, and $\times$ in Fig. 3, I). If neither player upgrades, the player on the left faces low *expected* cost of contamination due to the low probability of flow to the west, whereas the player on the right faces a high expected cost of contamination due high probability of flow to the east. This creates an asymmetric dilemma where the left household chooses not to upgrade in the Nash equilibrium, but would need to upgrade to reduce costs for the

other player and achieve the Social Optimum. Multiple types of Asymmetric Dilemmas exist. In AD-i, one player upgrades in the Nash equilibrium because they have a high risk of self-contamination. However the social optimum requires that the second player upgrade because both have a modest risk of contamination from this player. In this situation the Nash equilibrium is (E, B) or (B, E) and the social optimum is (E, E). In AD-ii, neither player faces a high risk of self-contamination, but one

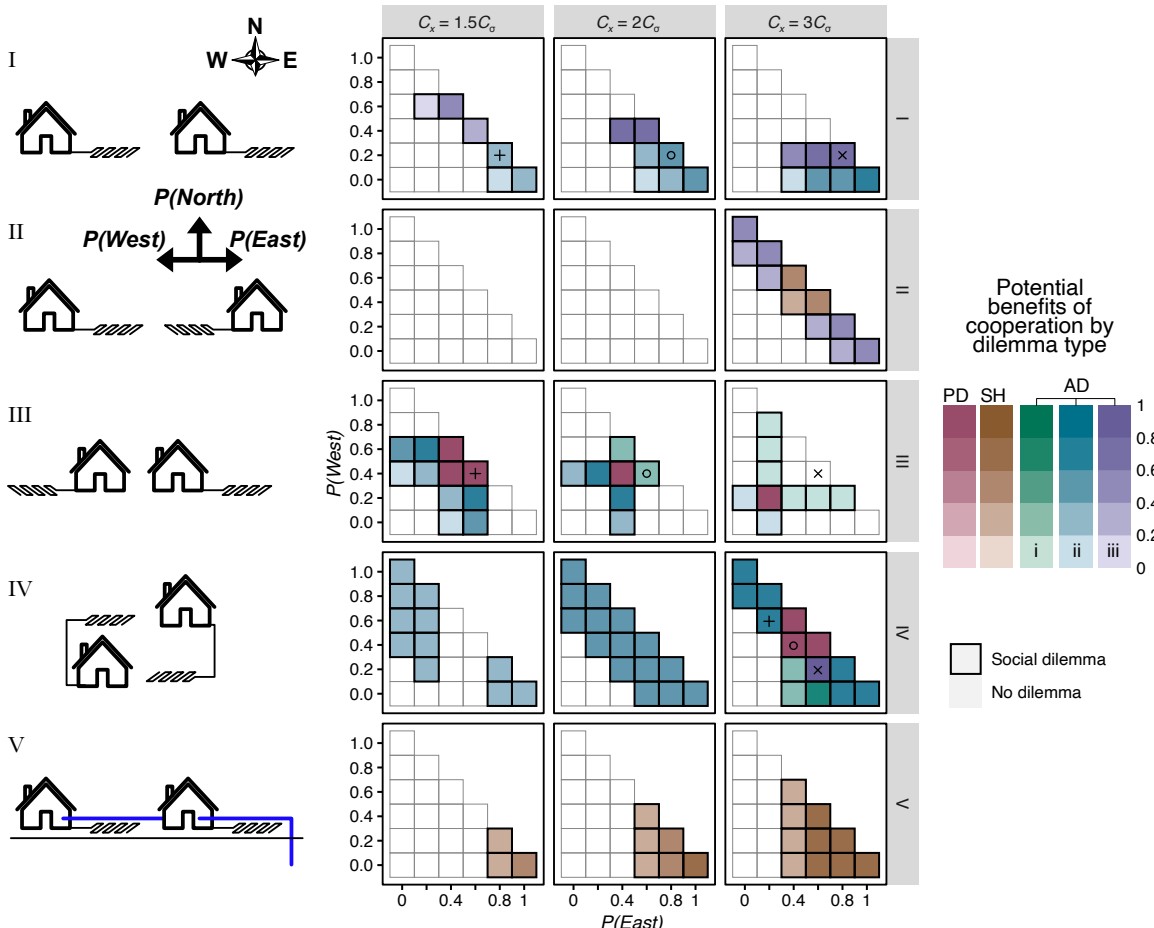

**Figure 3.** Social dilemmas under various household configurations and various cost ratios. (left) Household configurations I-V, with uncertainty about the mean direction of groundwater flow. (right) Social dilemma types Prisoner's Dilemma (PD), Stag-Hunt game (SH), and Asymmetric Dilemma (AD) along with the potential benefits of cooperation, given as the difference in normalized payouts between the social optimal and the (least desirable) Nash equilibrium, averaged for both players. The main axes represent the probability of flow towards the west and east, respectively, with the remainder being the probability of flow towards the north (i.e., $P(North) = 1 - P(West) - P(East)$).

player faces a high risk of contamination from the other player. The Nash is (B, B) and the social optimal is (E, B) or (B, E). Similarly in AD-iii, neither player has a high risk of self-contamination. However, the combined risk of contamination from either player means that both players must upgrade to achieve the Social Optimum. In this situation, the Nash is (B, B), and the social optimum is (E, E). Although the two NE and SO decisions are identical to those in the Prisoner's Dilemma, this situation is not a Prisoner's Dilemma because one player is still better off in the NE (B, B) than the SO where both players upgrade (E, E).

Generally, there tends to be greater likelihood of a social dilemma if the relative risk of contamination is high. This can arise if the cost of contamination is high compared to the cost of upgrading (i.e. $C_x >> C_\sigma$, Fig. 3, right) or if the probability of contaminating the other player is high compared to the probability of self-contamination. Conversely, a high probability of self-contamination circumvents social dilemmas by incentivizing the contaminating household to upgrade, as seen in scenario I ($P(West) \to 1$) and III ($P(West) \to 1$ or $P(East) \to 1$).

Changing costs (either $C_x$ or $C_\sigma$) lead to transitions across different types of social dilemmas, indicated by the symbols $+, \circ, \times$ in Fig. 3, with payouts shown in Fig. 4. For instance in configuration I, the social dilemma changes from AD-ii with $C_x = 2C_\sigma$ ($\circ$) to AD-iii with $C_x = 3C_\sigma$ ($\times$) (Fig. 4a). Conversely, in configuration III, the social dilemma changes from a Prisoner's Dilemma with $C_x = 1.5C_\sigma$ (+) to AD-i with $C_x = 2C_\sigma$ ($\circ$) to no dilemma with $C_x = 3C_\sigma$ ($\times$) (Fig. 4b).

Changes in the uncertainty associated with groundwater behavior can also lead to transitions across different types of social dilemmas. In particular, in configuration IV with $C_x = 3C_\sigma$, the social dilemma changes from AD-ii with $P(East) = 0.2$ (+) to PD with $P(East) = 0.4$ ($\circ$) to AD-iii with $P(East) = 0.6$ ($\times$) (Fig. 4c). These dynamics have practical implications because each type of dilemma (PD, SH, and AD) is associated with a distinct set of appropriate policy responses, as discussed in Sect. 5.

## 3   A general N-player game and groundwater model

### 3.1   Social dilemmas in the symmetric game

The two-by-two games described above demonstrate the variety of social dilemmas that can exist within groundwater contamination games. We now translate the game to an N-player system in which each player faces the same decision ($\sigma_i$) and costs ($C_x$ and $C_\sigma$) as above, while also generating a more plausible representation of groundwater behavior and probabilities of contamination.

Consider the binary random variable $X_i$, which denotes the event that player $i$ has their well contaminated. In this version of the game, the associated probability $p_i = P(X_i)$ is the union of probabilities that the well is contaminated by any septic system $j$:

$$p_i = \bigcup_j p_{ij}. \tag{2}$$

We write the probability of well $i$ being contaminated by septic tank $j$ as $p_{ij} = P(X_{ij} \mid \sigma_j, d_{ij}, G)$, where $d_{ij}$ is the distance between the domestic well and septic system, and $G$ represents the set of hydrogeological parameters that determine groundwater flow. The probability $p_{ij}$ therefore encapsulates the groundwater hydrology between $i$ and $j$ and the decision by $j$ whether or not to upgrade their septic system, given by $\sigma_j$. The probability $p_{ij}$ is associated with each of the N players and can be determined via groundwater modeling (Sect. 3.2).

We begin by considering the key economic trade-off that is encapsulated by the decision (and associated risks) of whether or not to upgrade a septic system. Utility within the game can be considered from the perspective of the individual household

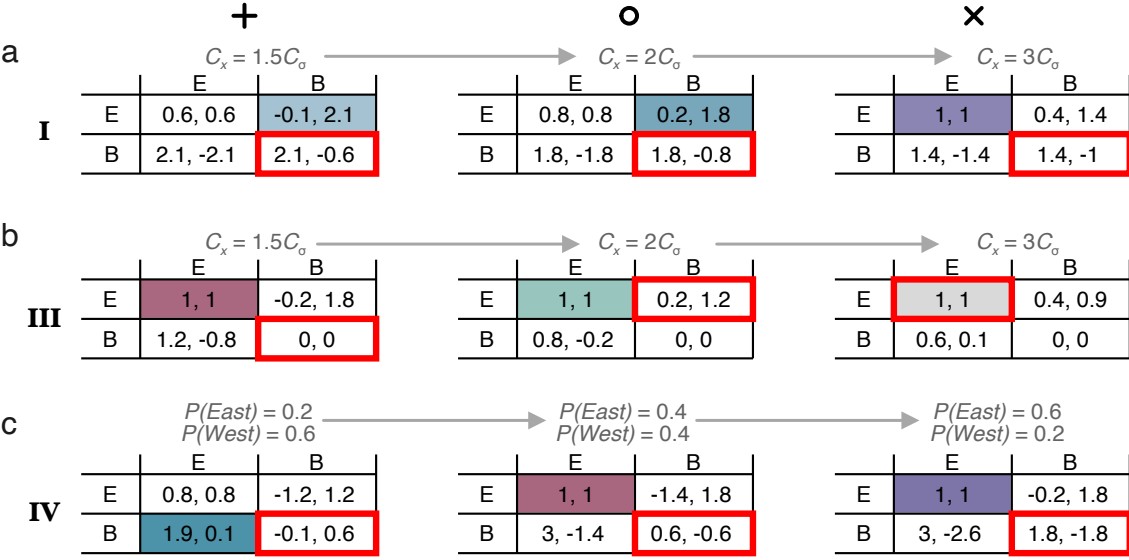

**Figure 4.** Transitions among social dilemmas by changing the cost ratio and the probability of groundwater flow direction. Each scenario corresponds to the configuration (I, III, IV) and parameters (+, ○, ×) presented in Fig. 3. (a) Household configuration I with $P(West) = 0.2$, $P(East) = 0.8$, and multiple cost ratios including $C_x = 1.5C_\sigma$ (AD-ii), $C_x = 2C_\sigma$ (AD-ii), and $C_x = 3C_\sigma$ (AD-iii). (b) Household configuration III with $P(West) = 0.4$, $P(East) = 0.6$, and cost ratios including $C_x = 1.5C_\sigma$ (Prisoner's Dilemma), $C_x = 2C_\sigma$ (AD-i), and $C_x = 3C_\sigma$ (no dilemma). (c) Household configuration IV with $P(West) = 0.2$, $C_x = 3C_\sigma$, and multiple probabilities flow direction including $P(East) = 0.2$, $P(West) = 0.6$ (AD-ii), $P(East) = 0.4$, $P(West) = 0.4$ (Prisoner's Dilemma), and $P(East) = 0.6$, $P(West) = 0.2$ (AD-iii).

(whether or not they should upgrade) and the perspective of the community (whether or not everyone should upgrade). Because the game is fully symmetric, we focus on the utility of an individual player given by Eq. (1), which also representative of the mean utility for households in the community. The reduction in probability of contamination for player $i$ if everyone upgrades is given as $\Delta p_i = p_i(\sigma_j = 0) - p_i(\sigma_j = 1) \forall j$. All players would be better off with everyone upgrading if $\Delta p_i C_x / C_\sigma - 1$ is positive. In contrast, the reduction in the probability of contamination of player $i$ if only player $i$ upgrades is $\Delta p_{ii} = p_i(\sigma_i = 0, \sigma_j = 0) - p_i(\sigma_i = 1, \sigma_j = 0) \forall j \neq i$. An individual player wishes to upgrade their septic system if $\Delta p_{ii} C_x / C_\sigma - 1$ is positive. Furthermore, a social dilemma occurs when

$$\Delta p_{ii} C_x - C_\sigma < 0 < \Delta p_i C_x - C_\sigma . \tag{3}$$

In other words, all players would benefit from upgrading (right hand side), but any individual player will endure a decrease in utility if they are the only one to upgrade (left hand side). The occurrence of a social dilemma can be understood by plotting these quantities against each other (Fig. 5). In particular, high $\Delta p_i$ and $C_x / C_\sigma$ increase the benefits of cooperation (everyone upgrades) because the consequences of not upgrading are steep (high probability, and associated costs, of contamination). Conversely, low $\Delta p_{ii}$ and $C_x / C_\sigma$ increase the maximum loss of upgrading individually (i.e., the expected decrease in utility

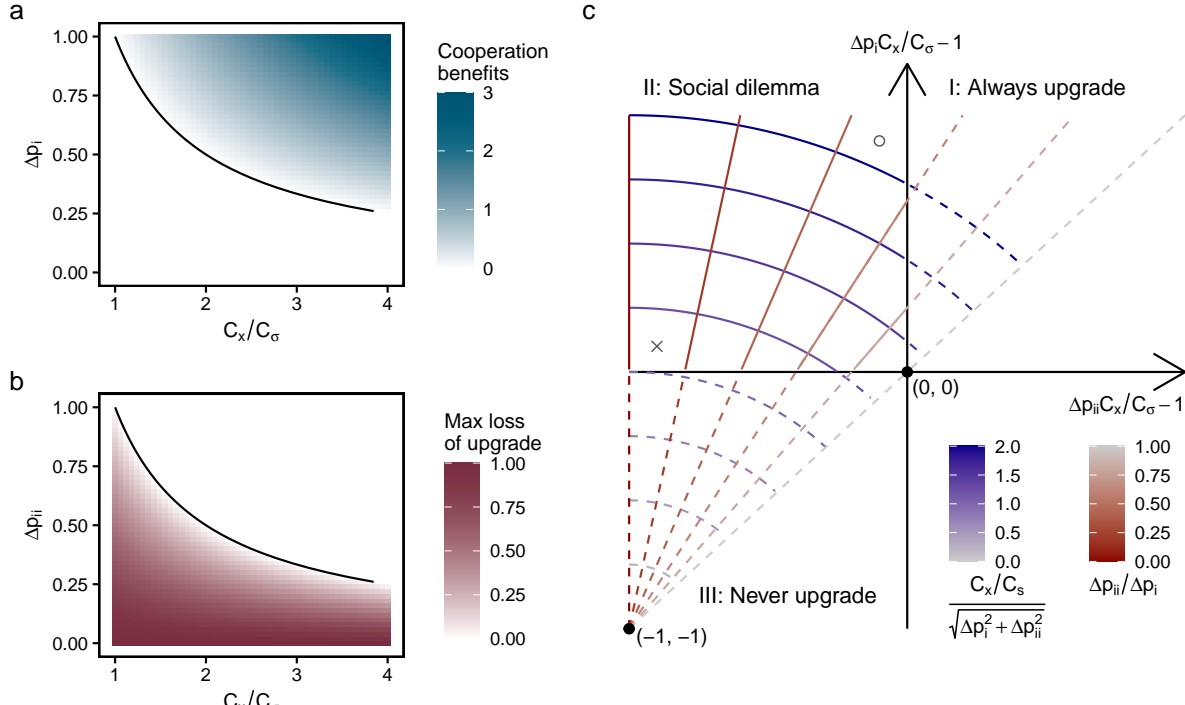

**Figure 5.** Social dilemmas as a result of individual and collective upgrades. (a) Benefits of cooperation, given as $\Delta p_i C_x / C_\sigma - 1$. (b) The maximum loss of upgrading, given as the expected decrease in utility for a single house to upgrade in the absence of cooperation $(1 - \Delta p_{ii} C_x / C_\sigma)$. (c) Benefits of cooperation versus the (inverse) maximum loss of upgrading, with a social dilemma occurring in quadrant II. Blue contours indicate constant cost ratios $(C_x / C_s)$, with higher cost ratios increasing the benefits of upgrading. Red contours indicate the ratio of $\Delta p_{ii} / \Delta p_i$, with lower values meaning that players get little benefit from upgrading their own septic tank relative to all other players upgrading.

when no one else upgrades), because individual benefits of upgrading are small. Both the cost ratio $C_x / C_\sigma$ and probability ratio
$\Delta p_{ii} / \Delta p_i$ play an important role in determining whether or not a social dilemma occurs, and characterizing social dilemmas requires understanding both properties together (Fig. 5c).

Cooperation is more likely to emerge when the maximum loss of upgrading (which occurs if a player is the only individual to upgrade) is low relative to the benefits of cooperation. This situation is represented by the ∘ in Fig. 5c, which could occur when the cost of contamination is relatively high ($C_x >> C_\sigma$) and there is a reasonable chance of self contamination ($\Delta p_{ii} / \Delta p_i \to 1$)
but not enough to convince an individual to upgrade without collective action (i.e., not in quadrant I). Conversely, cooperation is likely more difficult when the maximum loss of upgrading is high in comparison to the benefits of cooperation. This situation is represented by the × in Fig. 5c, and would occur when the cost ratio ($C_x / C_\sigma$) is just over 1 and the probability of self contamination is negligible ($\Delta p_{ii} \to 0$).

## 3.2 Groundwater model

Groundwater behavior determines the probability that any well $i$ could be contaminated by leachate from a septic field $j$, $p_{ij}$. These probabilities and their unions, as required by Eq. (3), could be determined via any groundwater model appropriate for the circumstances. For instance, Horn and Harter (2009) used a MODFLOW model with a fine-scale grid to demonstrate that domestic wells capture groundwater primarily from a region that approximates a cylinder extending outwards from the screened portion of the well. Bremer and Harter (2012) extended these conceptual findings by assuming constant recharge and uniform lateral groundwater flow, such that the probability of domestic well contamination could be determined purely from the configuration of septic tanks and domestic wells along with uncertainty on the mean direction of flow.

Based on these considerations, we develop a parsimonious groundwater model that incorporates uncertainty in the vertical direction of flow, allowing the calculation of $p_{ij}$ using neighborhood layouts and simple aquifer geometries. The model is shown in Fig. 6, and can be calibrated such that overall probability of contamination reflects actual contamination rates. In the model, the probability of contamination $p_{ij}$ is equivalent to the probability that a particle from septic system $j$ intersects well source area $i$ (Fig. 6). The model encapsulates uncertainty in the mean horizontal direction of flow ($\theta$, Fig. 6a) as well as uncertainty in the flow trajectory within the vertical plane ($u_z/u_r$, Fig. 6b), given that the particle may pass over or underneath the well (Horn and Harter, 2009).

Flow velocities in the $\theta$-$z$ plane are determined by the vertical and horizontal velocities, $u_z$ and $u_r$, which are, in turn, determined by the vertical and horizontal fluxes. We assume steady state behavior, such that the vertical flux is given by average annual seepage to the water table, $S$. All water is assumed to eventually flow toward a linear sink (e.g., a gaining stream), so that the horizontal flux can be estimated as the rate of seepage multiplied by the uphill contributing area. When considering a unit width of the aquifer, this rate is given by $S \times L$, where $L$ is the distance to the groundwater divide.

For parsimony and in the absence of additional information, we assume that uncertainty in the mean direction of groundwater flow can be represented by a uniform distribution, $\theta \sim U(\theta_1, \theta_2)$. Uncertainty in the $\theta$-$z$ trajectory of flow is determined by uncertainty in the distance to the groundwater divide, also parameterized as as uniform distribution, $L \sim U(L_1, L_2)$. Wells are cased from the ground surface to depth $z_1$ and screened from $z_1$ to $z_2$. Flow paths to the well occur only from groundwater within the range of depth of the screened portion of the well, $[z_1, z_2]$. The assumption of uniform uncertainty in flow direction has been applied in similar systems (Bremer and Harter, 2012), and the possibility that contamination can pass over or underneath a well has been demonstrated using MODFLOW (Horn and Harter, 2009). The groundwater model was implemented in an R package (Penny, 2021) that allows the calculation of both $p_{ij}$ and $p_i$. As information is gained and the most critical components identified, more complex representations can readily be added, but starting with a parsimonious representation and set of models has significant benefits with any probabilistic risk assessment (Bolster et al., 2009; de Barros et al., 2011).

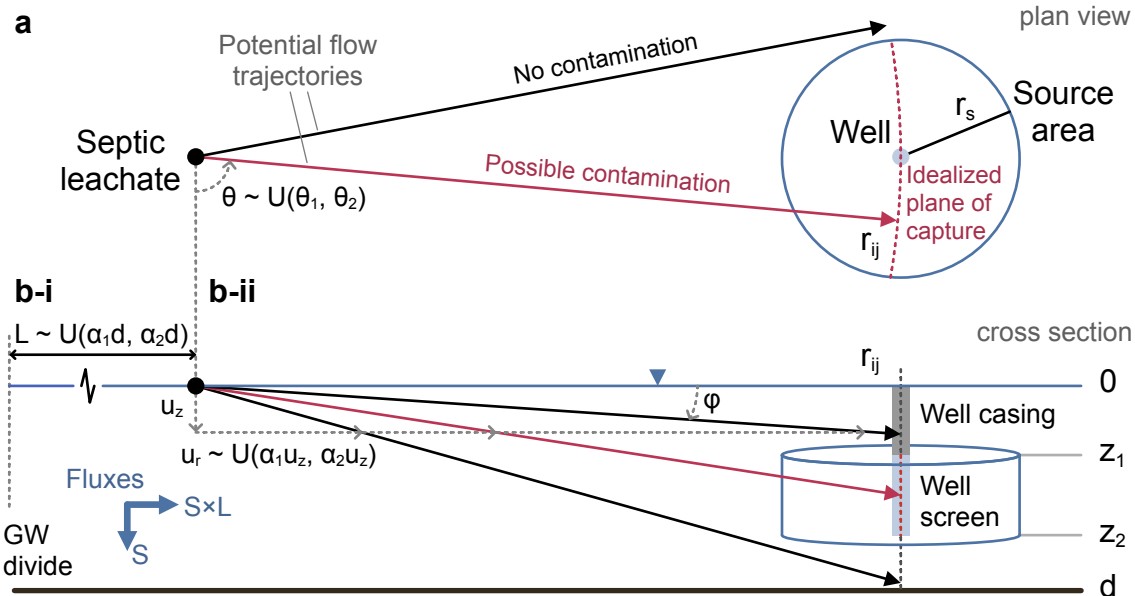

**Figure 6.** Groundwater model to determine the probability of contamination of well $i$ from septic system $j$. (a) Plan view of the septic field, domestic well, and possible flow trajectories. The mean direction of flow is unknown and follows a uniform distribution with $\theta \sim U(0, 2\pi)$. (b-i) Vertical and horizontal groundwater fluxes are determined by seepage, $S$, and the distance to the groundwater divide, approximated by $L \sim U(0, L_{max})$. (b-ii) The well is cased to a depth $z_1$ and screened from $z_1$ to $z_2$, with the possibility that contaminants can pass over or under the well. Velocities are determined from the fluxes such that $u_z = f(S)$ and $u_r = f(S \times L)$. It follows that $u_z$ is fixed while $u_r$ contains uncertainty represented by a uniform distribution. The well is contaminated if the flow path intersects the idealized plane of capture (red dashed line), representing the source area of the well, in both the horizontal and vertical planes. See text for details.

## 4   Case study

### 4.1   Study site: St. Joseph County, Indiana

In order to evaluate the potential for social dilemmas to emerge in real-world scenarios, we apply the N-player game described above to community nitrate contamination in St. Joseph County in northwestern Indiana, United States (see Fig. 1b). The county is home to 271,000 residents, with approximately half (152,000) living in the cities of South Bend and Mishawaka (U.S. Census Bureau, 2019). Outside the city boundaries, the most common water supply and treatment are domestic wells and septic systems, respectively, that abstract from and discharge to the St. Joseph Aquifer (Bayless and Arihood, 1996). We obtained data on nitrate levels in domestic wells from the St. Joseph County Health Department (https://www.sjcindiana.com/302/Health-Department), which maintains publicly available records of nitrate tests recorded when new wells are installed or when houses are sold. The data confirm that in higher-density residential areas of the county, nitrate contamination is common, particularly in Centre Township and Granger Census Designated Place (CDP). For instance, 7% and 5% of household tests,

in Centre Township and Granger, respectively, exhibited nitrate concentrations over the Environmental Protection Agency (EPA) Maximum Contaminant Level (MCL) of 10 ppm. Moreover, 43% and 30% of household tests in each of the two areas exhibited nitrate concentrations over 5 ppm. To address this problem, the city has been working towards extending sewer lines into Granger, but sewer connections are optional and residents must pay a fee for the connection (St. Joseph County, 2011). New developments may be more likely to be connected to piped water and sewers (Sheckler, 2021), but the issue in existing homes remains. As yet, there is no clear solution to the problem. To explore the possibility of social dilemmas and potential policy solutions, we generate parameter estimates for the contamination game described above, and calibrate the model to observed prevalence of contamination in Centre Township and Granger CDP.

## 4.2 Model parameterization and calibration

Both the economic portion of the model and the groundwater portion must be parameterized. For the economic portion, a complete cost of contamination is difficult to obtain. The health consequences of nitrate contamination are most acute for infants who may develop methaemoglobinaemia (Knobeloch et al., 2000), whereas the long-term consequences of exposure for children and adults are potentially concerning but unclear (Ward et al., 2018). We therefore focused on the perceived cost of contamination, using existing empirical studies that associate the effect of contamination with a reduction in the price of houses. In particular, Guignet et al. (2016) used a hedonic analysis of nitrate well tests and real estate transactions in Lake County, Florida to determine that the typical reduction in home value was 2-6% for those homes with nitrate contamination over the EPA MCL. We consider this loss in home value to be a plausible estimate of the cost of contamination.

The cost of upgrading the septic system to reduce or prevent nitrate contamination has been documented for EPA approved systems. We consider the Aquapoint, Inc., Bioclere™ 16/12 system, with a typical price range of $6000-8000 USD and Bio-Microbics, Inc., RetroFAST®0.375 System, with a price range of $4000-5500 USD (Washington State Department of Health, 2005; US EPA, 2007).

For the groundwater model, we note that we are interested in determining $p_i$, which is the probability of contamination of each well $i$ with respect to all septic systems. Assuming the configuration of wells and septic systems are known, this probability is determined primarily by uncertainty in groundwater flow paths and the capture region of domestic wells. We therefore specified the groundwater model as follows. We assumed that the mean direction of flow can be parameterized by a uniform distribution spanning West to South (in Granger) and Northwest to Northeast (in Centre), roughly approximating the most common directions of flow in the St. Joseph Aquifer towards the St. Joseph River (Bayless and Arihood, 1996). Indiana regulations require that wells are installed at a depth of at least 25 ft (7.6 m) (Indiana State Department of Health, 2021), and the depth to the groundwater table is typically 5-15 ft (Bayless and Arihood, 1996). Given that wells reduce the water level locally, we assume a groundwater depth at the upper end of this range (15 ft, 4.6 m). We further assume that wells were screened down to the bottom of this aquifer layer (70 ft, 21.3 m) (Bayless and Arihood, 1996). We approximate the distance to the groundwater divide as a uniform distribution characterized by the average width of the aquifer divided by the average height, giving an approximate ratio of 10-to-1 (Bayless and Arihood, 1996). Although these parameters are representative of

the groundwater scenario, the uncertainty embedded in these estimates underscores the importance of calibrating the model to observed contamination probabilities.

We use the radius of well capture ($r_s$) as the only calibration parameter. This value represents the maximum lateral distance from the well that the center of a septic plume could pass through to generate contamination within the well. Therefore, the value could potentially change for different regions of the aquifer and different thresholds of contamination. Note that the model should capture the perceived probability of contamination based on information available to the players, rather than reflect the most precise estimate based on complete knowledge of the groundwater system. As such, our stylized model and

calibration procedure associates this probability with household density and observed probabilities of contamination.

    St. Joseph County requires that water quality of private wells be tested any time a new well is installed or a property with a private well is sold. We obtained records of nitrate contamination from these tests for both Centre Township (N = 724) and Granger CDP (N = 3457) from the St. Joseph County Department of Health, spanning the years 2012–2019. After rasterizing the study region to 16-acre pixels, we determined the probability of contamination as the fraction of household tests in each

pixel that exceeded a particular contamination limit, and matched each pixel with an associated housing density from publicly available county shapefiles. Only residential and agricultural zoning areas were considered, and we determined the housing density of each pixel by counting the number of local address within the pixel using data obtained from the St. Joseph County GIS website (SJCGIS, 2020b, a). We then calibrated the groundwater model in both Centre Township and Granger CDP for contamination thresholds of 5 ppm and 10 ppm. The calibration procedure led to an estimate of the probability of contamination

based on housing density and the stylized groundwater model (Fig. 7a).

    In general, with a 10 ppm threshold, the probability of contamination is low (see figure 7a). With a 5 ppm threshold, the likelihood is much higher, and there is a clearer relationship between contamination rates and housing density. The issue of social dilemmas depends on assumptions of the system. We specify four scenarios, representing different assumptions on the costs and likelihood of contamination. In all cases the cost of contamination is a fraction of the home value:

– i. Centre 5 ppm calibration, $4000 to upgrade septic, $C_x$ equals 5% of home value.

    – ii. Centre 10 ppm, $4000 to upgrade septic, $C_x$ equals 5% of home value.

    – iii. Granger 5 ppm, $8000 to upgrade septic, $C_x$ equals 4% of home value.

    – iv. Granger 10 ppm, $8000 to upgrade septic, $C_x$ equals 4% of home value.

    We calibrated $r_s$ for each scenario using observed data, allowing us to determine $\Delta p_i$ and $\Delta p_{ii}$ based on housing density.

320     We used three values housing density to evaluate the model: 1, 2, and 3 households per acre. We further calculated the cost of contamination for multiple values of home prices: $100k, 400k, and 700k. These can be mapped to the space $\Delta p_i$ vs $C_x/C_\sigma$ (Fig. 7c).

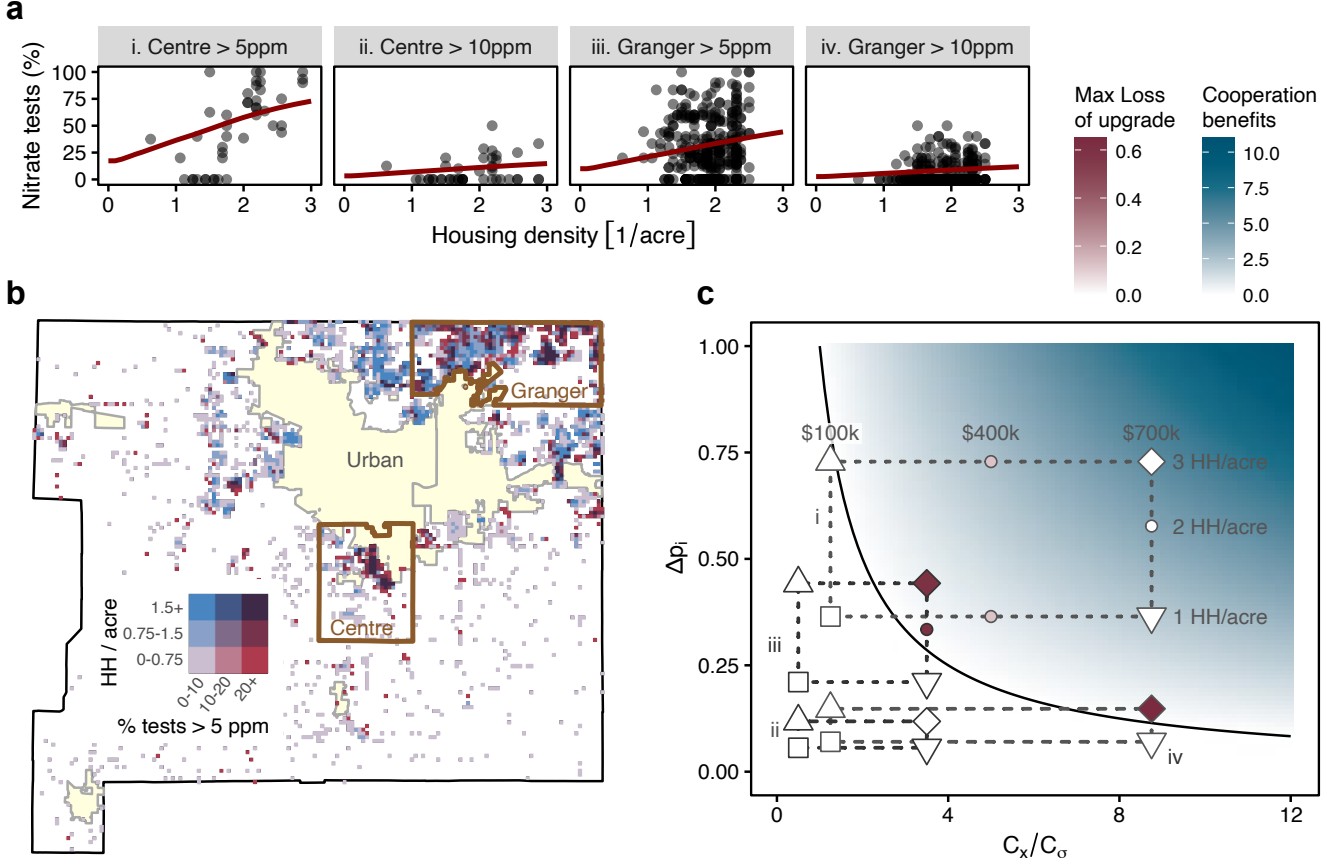

**Figure 7.** St. Joseph Case study. (a) Percent of nitrates tests exceeding a given threshold versus the housing density for Centre Township and Granger CDP. (b) Map of St. Joseph County, with colors indicating the housing density and percent of nitrates tests exceeding 5 ppm in 16-acre pixels. (c) Social dilemmas based on housing price and housing density (shown), and costs of contamination and septic upgrades (not shown, see text). Cooperation benefits, $\Delta p_i C_x / C_\sigma - 1$, are shown as background shading (in blue) and the maximum loss of upgrading, $1 - \Delta p_{ii} C_x / C_\sigma$, is shown as the fill color of example points (in red). Note that both are defined the same as in Fig. 5.

### 4.3 Social dilemmas

With a 10 ppm threshold, it is generally unlikely, except for more expensive houses (see Fig. 7c for ranges), that a social
dilemma will occur under the assumptions in scenarios ii and iv. With a 5 ppm threshold and assumptions as shown, it is
much more likely that a social dilemma would occur. In fact, in some cases everyone chooses to upgrade (no social dilemma).
This represents an extreme case with high cost of contamination (i.e., large $C_x / C_\sigma$) and co-located wells and septic systems
(positive $\Delta p_{ii}$). We note that if the direction of groundwater flow were known, the septic would likely be placed downstream
from the well such that $\Delta p_{ii}$ would be reduced and these situations could become social dilemmas. Additionally, the 5 ppm
threshold is below the EPA MCL for nitrates, and therefore $C_x / C_\sigma$ could be overestimated. However, this ratio captures only

the internalized costs to the homeowner and not additional public health or environmental externalities that could affect the socially optimal utility. This scenario is therefore illustrative of what could happen when such externalities are incorporated. Heterogeneity in groundwater contamination still plays a large role. For instance, in some neighborhoods the rates of contamination were as high as 50% of households over 10 ppm, whereas the calibrated model estimated a contamination rate of about 12% (Fig. 7a, ii and iv). Thus the model is conservative in some cases and, if these differences could be resolved, then social dilemmas could be evaluated with greater precision. For instance, some wells tap into the deep aquifer, but only 10% of wells in the dataset came with well depth, therefore it is possible that wells in the shallow aquifer have higher rates of contamination and the distinction is not captured within the dataset. Other factors in St. Joseph County may lead local abnormalities. For instance, septic design and maintenance practices play a role in septic water quality and well contamination, and lakes exist in some neighborhoods that may be recharging and diluting groundwater.

## 5  Discussion

The results demonstrate that multiple types of social dilemmas can arise in the context of groundwater contamination of private water supply and sanitation systems. We first describe theoretical interpretations of the two-player and N-player versions of the game, followed by policy considerations in regions such as St. Joseph County.

### 5.1  Theoretical implications

When considering two-player games with uncertainty in the mean direction of groundwater flow, we identified five types of social dilemmas including a Prisoner's Dilemma, Stag-Hunt game, and three types of Asymmetric Dilemmas. The type of social dilemma can serve as a guideline to understanding barriers to cooperation and policy opportunities to shift from the Nash equilibrium to the Social Optimum (Fig. 8).

The *Prisoner's dilemma* is, in essence, a free-rider problem: a market failure that occurs when people might benefit from services without having to pay the associated cost (Fischbacher and Gachter, 2010). Here, the decision keep a basic system (B) strictly dominates the decision to upgrade to an enhanced system (E), and each player benefits the most when only the other player upgrades. This allows the player in question to utilize the cleaner water associated with the other player's upgraded septic without having to pay for an upgrade themselves. A variety of market design approaches have been identified to address the free-rider problem in the context of public goods, including assurance contracts (Tabarrok, 1998), Coasian bargaining (Farrell, 1987) and incentives for cooperation (Balliet et al., 2011). The Nash equilibrium in the *Prisoner's dilemma* game is not Pareto efficient, meaning that all players can benefit if everyone upgrades their septic system. The free-rider problem can therefore also be addressed through altruistic social norms (potentially combined with enforcement and sanctions) to support collective action (Kerr, 1992). In other words, despite the various obstacles to overcoming the Prisoner's Dilemma, there exist multiple governance approaches to address this challenge and initiate community-wide upgrades to enhanced septic systems, allowing everyone to benefit from cleaner groundwater and reduced contamination costs.

**Figure 8.** Pareto efficiency and upgrades required for the Social Optimum. The five types of dilemmas can be mapped according to whether or not there is a Nash equilibrium that is Pareto *inefficient*, and the number of players that would have to upgrade to achieve a social optimal outcome (PD and AD games). In the Stag-Hunt game, if one player can be convinced to upgrade then the other player will be incentivized to upgrade as well.

In contrast, the *Stag-Hunt* game (also known as assurance game) is a problem of strategic uncertainty, where players might fail to achieve a socially-optimal equilibrium by not being able to coordinate their decision-making. Here, either player can only benefit from upgrading if the other player upgrades as well. Under these conditions, they will only endure the cost of
365 upgrading if they are reasonably sure that the other player is also upgrading. Strategic uncertainty can be resolved through trust and reliable information (Jansson and Eriksson, 2015). In contrast with the Prisoner's Dilemma, where *all* players must be incentivized to upgrade, the assurance game only requires *any* player to buy in (e.g., through targeted subsidies). The other player will reciprocate if she (he) has sufficient assurance of the other player's buy in. As such, public programs that provide some measure of transparency or assurance that some households have upgraded their septic systems would encourage
additional households to also upgrade.

In each of the *Asymmetric dilemmas*, the Nash equilibrium is Pareto optimal but one player receives outsize benefits at the expense of the other player. A greater social optimum can be achieved by one player sacrificing some of their utility. Asymmetric Dilemmas i and ii are akin to the Stag-Hunt game in that the social optimal is one upgrade away from the Nash equilibrium. However, unlike the Stag-Hunt game, achieving the social optimal requires convincing a *specific* player to
375 upgrade. Any subsidy must therefore specifically targeted to the appropriate player, which requires reliable information on all players' utility parameters. Finally, Asymmetric Dilemma iii is partially similar to a Prisoner's dilemma because the social optimal requires both players to upgrade. However, unlike the Prisoner's dilemma where both players increase their utility by achieving the Social Optimal, one player is worse off in the social optimal compared to the Nash Equilibrium. In this situation, targeted subsidies or side payments might be needed, in addition to social or market approaches to address potential free-rider
problems.

Social dilemmas in N-player games can be understood by non-dimensionalizing the expected utility when everyone upgrades (i.e., the cooperation benefits) and expected utility when only one player upgrades (i.e., the maximum loss of upgrading). Social dilemmas arise when both are positive. Situations akin to the Prisoner's dilemma are likely to arise when the cooperation benefits are modest and the maximum loss of upgrading is high. Dilemmas akin to a Stag-Hunt game are more likely when the cooperation benefits are high and the maximum loss of upgrading is low and $p_{ii} \cap p_{ij} > 0$, meaning that an individual's upgrade becomes more appealing after another player upgrades. The three asymmetric dilemmas only emerge in situations where (known) specific players have undue influence on the contamination of other players' wells. We assume that the N-players are homogeneously located throughout the neighborhood and do not have perfect information on the flow direction of groundwater, and that all households face similar costs and emit comparable pollution loads. Under these conditions, asymmetric dilemmas would only emerge near the edges of the neighborhood and therefore, assuming the neighborhood is sufficiently large, only concern a small fraction of the households. Although both types of symmetric social dilemmas (Prisoner's Dilemma and Stag-Hunt) are possible, the Prisoner's dilemma situation is more likely because the case study in St. Joseph County suggests that our model potentially overestimates $\Delta p_{ii}$ and the maximum loss of upgrading (Fig. 4). However, the specific benefits and costs of such upgrades are likely to be location-specific and depend on a variety of factors. While we show that such dilemmas are possible in St. Joseph County, additional groundwork would be needed to more clearly identify the type of situation that exists within the county, including the specific locations where housing and groundwater behavior give rise to such dilemmas.

### 5.2 Policy considerations

The model identifies how social dilemmas arise through tension between the cost of well contamination and inability of individual households to prevent contamination. In particular, wealthier households may be able to collectively organize to prevent contamination through enhanced septic treatment. This approach can be facilitated by homeowners associations, and public education about the consequences of contamination. As the model demonstrates, the drawback of this framing is that lower-income households may be averse to participation in such projects because the costs of enhanced septic treatment exceed the economic benefits associated with home prices. This creates an obvious conundrum for local governments whereby the most equitable health outcomes cannot be achieved through community collective action because lower-income households are reluctant to participate in collective initiatives for enhanced septic treatment. It is further problematic in the U.S. where low-income households are less likely to be supplied by piped water (Meehan et al., 2020b) and often face greater environmental health risks (Meehan et al., 2020a) than high-income households.

We can consider whether or not this possibility may be occurring in St. Joseph county by comparing Granger CDP and Centre Township. As both regions are located within unincorporated areas of St. Joseph County, they face the same regulations with respect to housing density and well placement. We obtained income distributions for each location from http:\censusreporter. org, which aggregates census statistics by geographical areas and provides median income and four income groups ($0-50k, $50k-100k, $100k-200k, >$200k). Centre Twp has a median household income of $65k and the most common income bracket is $0-50k (39% of households) (Census Reporter, 2022a), whereas Granger CDP has a median income of $102k and the most common income group is $100k-200k (35% of households) (Census Reporter, 2022b). In Centre Twp, nearly all dense

neighborhoods with over two houses per acre have at least 50% prevalence of contamination over 5 ppm (Fig. 7a-i). Conversely most dense neighborhoods in Granger with over two houses per acre have less than 50% prevalence of nitrate contamination over 5 ppm (Fig. 7a-iii). Thus, the relative prevalence of contamination in these two locations is consistent with the national trend where lower-income households face greater risks.

The above analysis suggests three potential policy approaches depending on the disposition of local residents. First, promoting homeowners associations that require enhanced septic treatment would be acceptable to wealthier households provided this initiative is combined with sufficient public education. Lower-income households may oppose such requirements because the economic burden of installing enhanced treatment likely exceeds the perceived benefits. Second, local governments can require wastewater treatment, either via enhanced septic systems or public sewerage. This option will likely lead to more equitable public health outcomes but potentially inequitable economic outcomes that disadvantage lower-income households who rent or own property with lower values. Third, the local government can leverage a combination of taxes and fees to incentivize (i.e., subsidize) wastewater treatment via enhanced septic systems or community sewerage. The game theoretic model provides a first estimate of the subsidies that would be required such that the cost of treatment would match the reduction in property value from nitrate contamination. Public outreach could be used to refine the taxation structure and value of subsidies so that residents are amenable to this approach.

## 5.3 Model assumptions and broader applications

This approach must take into consideration the assumptions and context in which the model was applied in St. Joseph County, including that (a) houses were uniformly distributed on a grid and (b) contamination occurred due to nitrate pollution and the cost of contamination was determined as the associated reduction in home prices.

The assumption of homogeneity yields situations of symmetry where the payouts are identical for all players. There are two features of heterogeneity that could affect model outcomes and therefore policy considerations. First, heterogeneity in the hydrogeology or housing density would create situations where some households are located upstream of the majority of the neighborhood and therefore would be less interested in contributing to collective action schemes that required enhanced septic treatment, as these households would be unlikely to be contaminated regardless of the behavior of other households. However, these theoretical limitations are unlikely to affect the policy process for the following reasons. To our knowledge, the majority of situations with high community nitrate contamination occur in hydrogeological settings with unconsolidated aquifers and flat topography, meaning that the direction of groundwater flow may not be obvious. It would therefore place local governments in a tenuous position to attempt to adequately assess the groundwater flow paths in order to customize policy to such asymmetries, and although they may exist they can be effectively ignored in most situations. Second, heterogeneity in property values would create a dynamic environment where wealthier households are willing to pay more than less wealthy households for improved water quality. This should be taken into account when implementing policy, and may facilitate implementation of the third policy approach, described above.

The cost of contamination could just as well arise from any range of household contaminants. We focused on nitrate contamination, which is prevalent in St. Joseph County and many other locations in the United States, but other situations may

arise with other types of pollutants and treatment strategies. The game can just as well be applied to these scenarios, account-
ing for different costs of contamination and treatment. For instance, pathogenic contamination is more prevalent in low- and
middle-income countries, and the associated economic consequences include loss of work and income (Prüss-Üstün et al.,
2016; Ngasala et al., 2019). Similarly, treatment strategies are likely to differ, both in terms of domestic water treatment and
waste treatment. A complete assessment is beyond the scope of this manuscript, but the game theoretic model could readily be
applied to such situations using the R package developed for this manuscript (Penny, 2021).

## 6  Conclusions

Contamination of domestic wells by septic leachate is a public health concern in residential areas with high housing density
around the world. The issue has been particularly difficult to resolve for public health officials and residents alike. We develop
a theoretical framework to understand how misaligned incentives can give rise to social dilemmas within such household water
systems. A variety of social dilemmas can occur in such situations, depending on the groundwater hydrology, costs of various
treatment technologies, and configurations of households. These dilemmas include the classic Prisoner's Dilemma and Stag-
Hunt game, as well as an Asymmetric Dilemma that favors some players over others. These dilemmas can occur even if all
households face the same economic costs of contamination and treatment. As such, this manuscript provides a theoretical basis
to identify when social dilemmas could be occurring and offers a new perspective through which to approach management of
such systems.

We applied the model to St. Joseph County, Indiana, in the U.S., and demonstrated that the occurrence of social dilemmas
depends on housing density and the perceived cost of contamination. The model indicates that wealthier households would be
more likely to organize via collective action to protect water quality, while poorer households would be reluctant to face costs
of upgrading, a discrepancy that makes equitable policy more difficult to achieve. Three possible options include community
education and organizing, treatment requirements, and subsidies for enhanced treatment. The most equitable policy option is
likely to be one that includes incentives to upgrade septic treatment for poorer households.

*Code and data availability.*  Data for domestic well densities in the U.S. were obtained from (Johnson and Belitz, 2019, https://doi.org/10.5066/P9FSLU3B
St. Joseph County GIS data were obtained from the SJC-GIS county mapping website https://www.sjcindiana.com/1743/GIS--County-Mapping.
The specific datasets accessed were Zoning Boundaries (SJCGIS, 2020b, https://sjcgis-stjocogis.hub.arcgis.com/datasets/zoning-boundaries)
and local addresses (SJCGIS, 2020a, https://sjcgis-stjocogis.hub.arcgis.com/datasets/address). The household nitrates data used in this article
were obtained from the St. Joseph County, IN, Public Health Department, and Requests of Public Documents for environmental health can
be made at https://www.in.gov/localhealth/stjosephcounty/environmental-health/. Code for the two-player games and groundwater model is
available in an R package (Penny, 2021) and can be accessed online at https://github.com/gopalpenny/nitratesgame.

*Author contributions.*  G.P., D.B., and M.F.M. designed the research. G.P. conducted the analyses. G.P. and M.F.M. wrote the manuscript.

*Competing interests.* The authors declare that they have no conflict of interest

*Acknowledgements.* The authors thank Brett Peters and Casey Stoffel for cleaning and preparing data on nitrates contamination in St. Joseph County, and Michèle Müller-Itten, Connor Mullen, and Bruce Huber for constructive feedback throughout the research process. The authors also acknowledge support from the National Science Foundation under Grant No. ICER 1824951 and SCC 1831669.

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
