# Peer review of "Social dilemmas and poor water quality in household water systems"

_Hydrology and Earth System Sciences, 2021_

## Author Response (AR1)

Dear Dr. Ursino,

Please find attached the revised version of our manuscript, now titled "Social dilemmas and poor water quality in household water systems", for consideration as a research article in HESS. We have carefully addressed comments by both of the reviewers, in particular the request to provide greater emphasis on the case study. With this in mind, I would like to highlight the following revisions to the manuscript:

- We have organized the Discussion section into subsections, and have included a deeper analysis of policy implications for the St. Joseph County case study. The two other subsections focus on theoretical implications of the model (from the original manuscript) and modeling assumptions and applications to different regions (new).
- We have revised the title based on a suggestion from Reviewer 2 that we refer to "household water systems" instead of "private water systems". We note that the previous title was "Social dilemmas and poor water quality in *private water systems*" (see revised title above).

We would like to thank the reviewers for their constructive comments and thank you for the opportunity to submit minor revisions. We have attached the revised manuscript, as well as a marked up version with tracked changes, both of which include my new affiliation appended to the title section. We include, below, a point-by-point response to reviewer comments (with our response in blue, and manuscript text in red).

Sincerely,

Gopal Penny (on behalf of the authors)

https://hess.copernicus.org/preprints/hess-2021-312/#discussion

**Comments from Reviewer 1**

Summary

In this paper, the author stressed a very important water issue. The private systems and wells contamination issued by septic systems, and this might cause many types of social dilemmas. The author. Has developed a water game theory for private water systems starting with 2 and then with N-players applied then a symmetric as well as an asymmetric game with more uncertainty related mainly with the groundwater flow direction and the different possibility of contamination. All this has been applied to learn more about the interlinkages between the choice of the septic system upgrading, wells contamination, and social dilemmas. To do so, the author applied a groundwater model. To have more clearance about. The different probabilities are related to the uncertainty around the groundwater flow direction and the possibility of contamination (self-contamination and cross-contamination) by the septic systems.

At the end of the game (study), we expect to have a straightforward decision (policy solutions) that can help in solving this social dilemma with less cost.

General comments:

1) Title: reflect very well the main question of the paper

Thank you. Please note that we intend to revise the title to "Social dilemmas and poor water quality in household water systems" in response to a comment from the other reviewer.

2) Introduction: well, framed, and organized, but the author needs to check for more recent references.

Thank you for this suggestion. We will add a number of recent references throughout the introduction and manuscript. These include papers referring to disparities in water quality (Schaider et al, 2019; Meehan et al, 2020a,b), household water contamination in developing countries (Ngasala, 2019), nitrate sources (Bastani and Harter, 2019) and source identification (Zendehbad, 2019), water quality hazard identification (Li et al, 2019; Juntaku et al, 2020), health risks, and temporal variation of water quality in domestic wells (Ornelas Van Horne et al, 2019).

Bastani, M., & Harter, T. (2019). Source area management practices as remediation tool to address groundwater nitrate pollution in drinking supply wells. Journal of contaminant hydrology, 226, 103521.

Juntakut, P., Haacker, E. M., Snow, D. D., & Ray, C. (2020). Risk and cost assessment of nitrate contamination in domestic wells. Water, 12(2), 428.

Li, P., He, X., & Guo, W. (2019). Spatial groundwater quality and potential health risks due to nitrate ingestion through drinking water: a case study in Yan'an City on the Loess Plateau of northwest China. Human and ecological risk assessment: an international journal, 25(1-2), 11-31.

Meehan, K., Jepson, W., Harris, L. M., Wutich, A., Beresford, M., Fencl, A., ... & Young, S. (2020a). Exposing the myths of household water insecurity in the global north: A critical review. Wiley Interdisciplinary Reviews: Water, 7(6), e1486.

Meehan, K., Jurjevich, J. R., Chun, N. M., & Sherrill, J. (2020b). Geographies of insecure water access and the housing–water nexus in US cities. Proceedings of the National Academy of Sciences, 117(46), 28700-28707.

Ngasala, T. M., Masten, S. J., & Phanikumar, M. S. (2019). Impact of domestic wells and hydrogeologic setting on water quality in peri-urban Dar Es Salaam, Tanzania. Science of the total environment, 686, 1238-1250.

Ornelas Van Horne, Y., Parks, J., Tran, T., Abrell, L., Reynolds, K. A., & Beamer, P. I. (2019). Seasonal variation of water quality in unregulated domestic wells. International journal of environmental research and public health, 16(9), 1569.

Schaider, L. A., Swetschinski, L., Campbell, C., & Rudel, R. A. (2019). Environmental justice and drinking water quality: are there socioeconomic disparities in nitrate levels in US drinking water?. Environmental Health, 18(1), 1-15.

Zendehbad, M., Cepuder, P., Loiskandl, W., & Stumpp, C. (2019). Source identification of nitrate contamination in the urban aquifer of Mashhad, Iran. Journal of Hydrology: Regional Studies, 25, 100618.

3) Two household contamination games: this part is very well described.

- However, there is a need to clarify if the game is cooperative or not cooperative before heading to the Nash equilibrium (non-cooperative).
- The author needs to explain why he has chosen to go with the non-cooperative choice, even if it was very clear that the author wanted to stress out the social dilemma cause. But it would be great to clarify the rationality behind the choice.
- Why the author did not consider all the parts of the game: elements of action (finite or infinite), information set (complete information or incomplete information game), numbers of the same play in a game (one-shot game and repeated game).

Thank you. We have rewritten the text to make explicit our rationale behind the setup of the two-household game. This includes that the game is a non-cooperative, static (one-stage of

play), one-shot (not repeated), game of complete information (the payout structure of all players is common knowledge). Note that the action space is limited to two actions (upgrade or do not upgrade). We use a one-shot game because the decision to build infrastructure locks players into future behavior given the capital cost and long lifespan of the infrastructure. These details will be clarified in the revised text as (L85):

We first consider a two-player static game with complete information, wherein the payout structure of both players is common knowledge. Note that we employ non-cooperative game theory to determine the presence or absence of social dilemmas, rather than the manner in which players build coalitions (as in cooperative game theory). As described below, this approach is realized by comparing outcomes under individual optimization (the Nash equilibrium) and full cooperation (the social optimum). Because the payout structure for all players is common knowledge, it is a game of complete information. In the game, each player chooses whether to upgrade their septic system to an enhanced system with contaminant removal (E) at some cost, or to keep a basic septic system (B) at no cost. The cost of upgrading, $C_\sigma$, represents the difference between the enhanced septic system and the cost for a conventional system (e.g., as mandated by local regulations). If a player's well becomes contaminated, that player also incurs a cost, $C_x$. Although the game can be played at any time, it is a static one-shot game because the action to upgrade a septic system only occurs once via capital investment and multiple stages (as in a dynamic game) are not necessary to determine the possibility of a social dilemma.

And just below (L98):

$\sigma_i \in \{0,1\}$ represents the action space of each household, which can choose to maintain a basic septic system (B, $\sigma_i = 0$) or upgrade to an enhanced system (E, $\sigma_i = 1$).

4) Symmetric two-player games: from lines 113 to 121: the paragraph is a bit complicated, there is clear contradiction and redundancy in explaining the upgrading and non-upgrading choice.

This paragraph describes Figure 2c and 2d, which show 2x2 payout structures and equilibria (both Nash and social optimum) for the two-player game. The two games (in Fig 2c and 2d) are similar but with key differences. We have modified the paragraph to address this comment while striving to seek a balance between clarity, brevity, and redundancy. The revised text is (126):

A social dilemma occurs when individual optimization leads to a different outcome than would be achieved if players work together. This can only occur when players potentially contaminate each other (as in Fig. 2c and 2d). When both players only contaminate the other player's well without contaminating their own well (Fig. 2c), the best response for each player would be not to upgrade (B), regardless of the decision of the other player. While neither player directly gains by upgrading their own septic system, each player would benefit if the other player upgrades. This leads to a classic Prisoner's dilemma (see Kollock, 1998) where neither player upgrades in the Nash equilibrium despite the mutual benefits of

doing so. In the final situation (Fig. 2d), players contaminate both their own and the other players' well. This game yields two Nash Equilibria, including one where neither player upgrades (B, B) and another where both players upgrade (E, E). This structure follows an Assurance or Stag-Hunt game (Kollock, 1998), where players only wish to upgrade to an enhanced system when the other player also upgrades. The first Nash Equilibrium (B,B) is not a Social Optimum and therefore creates a social dilemma.

5) Groundwater model: the author mentioned that the modeling part could be determined by any groundwater model, but he did not explain the rationale behind selecting /using the MODFLOW model.

We will revise this section to clarify that we do *not* use a MODFLOW model, but rather a simplified groundwater model based on the analytical element method. Our intention behind mentioning MODFLOW is to be clear that the non-cooperative game described in this paper can be implemented with *any* groundwater model that generates probabilities of contamination. We clarify in the manuscript (L227).

Based on these considerations, we develop a parsimonious groundwater model that incorporates uncertainty in the vertical direction of flow, allowing the calculation of pij using neighborhood layouts and simple aquifer geometries. The model is shown in Fig. 6.

6) Case study:

- The existing groundwater data are mainly based on assumptions, does the author performed any data collection or had access to any national database?
- It is important to have the year of any collected data
- Does wells capture radius (rs) is sufficient to calibrate a groundwater model? In my knowledge in the case of groundwater flow modeling we need more than the Rs parameter, we need for example the recharge and hydraulic conductivity supported by field data.
- The model validation is absent, does the author validate the model data?
- The author took too much space to explain the game (almost 10 pages), however, he didn't well calibrate and validate his groundwater model and he didn't clearly explain and apply the game for the case study.

The data were collected from the St. Joseph County Public Health Department, in the years (2012-2019). We realize this information was only mentioned at the beginning of the case study and not where we described data preparation and calibration. We agree this needs to be clearer, and we now explicitly describe the testing data and preprocessing in the manuscript in conjunction with calibration.

Although groundwater behavior and flowpaths can be extraordinarily complex, the intention of the groundwater model is not a complete representation of groundwater flow, but rather an estimate of the probabilities of contamination of any domestic well due to pollution from any septic system, as required by the game theoretic model. This probability depends only

on whether or not a pollutant will be captured by the well and (importantly) should capture the perceptions of the players. Under steady state and the simplified assumptions of our model, the capture radius and direction of flow are the key determinants of this probability. We assume the probability of flow direction is fixed, and therefore the well capture radius is the only calibration parameter. Note that the capture radius is the result of multiple hydrogeological features of the aquifer, including recharge rates, hydraulic conductivity, and well depth.

There are, of course, many other hydrogeological features that would determine whether or not a pollutant would be captured by a well. However, it is worth restating: in this paper we are *not* interested in a precise estimate that would determine whether or not a pollutant enters the well, but rather the appropriate estimate on the probability *based on the information available* to the players. In general, that information would be very limited because hydrogeological surveys and modeling are expensive. This feature of our study leads us to the simple model and calibration approach presented.

We have revised the manuscript text to read (L295):

We use the radius of well capture ($r\_s$) as the only calibration parameter. This value represents the maximum lateral distance from the well that the center of a septic plume could pass through to generate contamination within the well and is the result of multiple hydrogeological features including recharge rates, hydraulic conductivity, and well depth. Therefore, the value could potentially change for different regions of the aquifer and different thresholds of contamination. Note that the model should capture the perceived probability of contamination based on information available to the players, rather than reflect the most precise estimate based on complete knowledge of the groundwater system. As such, our stylized model and calibration procedure associates this probability with household density and observed probabilities of contamination.

St. Joseph County requires that water quality of private wells be tested any time a new well is installed or a property with a private well is sold (St. Joseph County, 2020). We obtained records of nitrate contamination from these tests for both Centre Township (N = 724) and Granger CDP (N = 3457) from the St. Joseph County Department of Health, spanning the years 2012--2019. From these tests, we determined the probability of contamination as the fraction of household tests in 16-acre pixels that exceeded a particular contamination limit, and matched each pixel with an associated housing density from publicly available county shapefiles. Households were determined as any local address within residential zoning areas, with data obtained from the St. Joseph County GIS website (SJCGIS2020a,b). We then calibrated the groundwater model in both Centre Township and Granger CDP for contamination thresholds of 5 ppm and 10 ppm. The calibration procedure led to an estimate of the probability of contamination based on housing density and the stylized groundwater model (Fig. 7a).

St. Joseph County, IND, Code of Ordinances, Ch. 52: Water Regulations, § 52.011 (2020)

7) Discussion: the discussion is good, but it is too general and does not directly reflect the results from the selected case study. The author here only explained more about the different types of social dilemmas.

Thank you for this comment, we will add to the discussion the following text to better focus on the case study and associated policy considerations (L398):

The model identifies how social dilemmas arise through tension between the cost of well contamination and inability of individual households to prevent contamination. In particular, wealthier households may be able to collectively organize to prevent contamination through enhanced septic treatment. This approach can be facilitated by homeowners associations, and public education about the consequences of contamination. As the model demonstrates, the drawback of this framing is that lower-income households may be averse to participation in such projects because the costs of enhanced septic treatment exceed the economic benefits associated with home prices. This creates an obvious conundrum for local governments whereby the most equitable health outcomes cannot be achieved through community collective action because lower-income households are reluctant to participate in collective initiatives for enhanced septic treatment. It is further problematic in the U.S. where low-income households are less likely to be supplied by piped water (Meehan, 2020a) and more face greater environmental health risks (Meehan, 2020b) than high-income households.

We can consider whether or not this possibility may be occurring in St. Joseph county by comparing Granger CDP and Centre Township. As both regions are located within unincorporated areas of St. Joseph County, they face the same regulations with respect to housing density and well placement. We obtained income distributions for each location from http:\censusreporter.org, which aggregates census statistics by geographical areas and provides median income and four income groups ($0-50k,$50k-100k, $100k-200k, >$200k). Centre Twp has a median household income of $65k and the most common income bracket is $0-50k (39% of households) (Census Reporter, 2022a), whereas Granger CDP has a median income of $102k and the most common income group is $100k-200k (35% of households) (Census Reporter, 2022b). In Centre Twp, nearly all dense neighborhoods with over two houses per acre have at least 50% prevalence of contamination over 5 ppm (Fig. 7a-i). Conversely most dense neighborhoods in Granger with over two houses per acre have less than 50% prevalence of nitrate contamination over 5 ppm (Fig. 7a-iii). Thus, the relative prevalence of contamination in these two locations is consistent with the national trend where lower-income households face greater risks.

The above analysis suggests three potential policy approaches depending on the disposition of local residents. First, promoting homeowners associations that require enhanced septic treatment would be acceptable to wealthier households provided this initiative is combined with sufficient public education. Lower-income households may oppose such requirements because the economic burden of installing enhanced treatment likely exceeds the perceived benefits. Second, local governments can require wastewater treatment, either via enhanced septic systems or public sewerage. This option will likely lead to more equitable public health outcomes but potentially inequitable economic outcomes that disadvantage lower-income households who rent or own property with lower

values. Third, the local government can leverage a combination of taxes and fees to incentivize (i.e., subsidize) wastewater treatment via enhanced septic systems or community sewerage. The game theoretic model provides a first estimate of the subsidies that would be required such that the cost of treatment would match the reduction in property value from nitrate contamination. Public outreach could be used to refine the taxation structure and value of subsidies so that residents are amenable to this approach.

8) Conclusion: we expected after applying the game to a case study to have some applicable policy recommendations/solutions, but the author didn't provide any straightforward solutions.

We have added policy recommendations in response to comment #7. We now briefly summarize these recommendations within the conclusions.

Specific comments:

1) Symmetric two-player games: from the line 113 to 121: the paragraph is a bit complicated, there is clear contradiction and redundancy in explaining the upgrading and non-upgrading choice

See response to #4, above.

2) Line 337: The other player will reciprocate if she (he) has sufficient assurance of the other player's buy-in.

We have made this correction.

3) I think that the references listed below worth to be sited in this publication:

- Raquel S, Ferenc S, Emery C Jr, Abraham R. Application of game theory for a groundwater conflict in Mexico. J Environ Manage. 2007 Sep;84(4):560-71. DOI: 10.1016/j.jenvman.2006.07.011. Epub 2006 Sep 22. PMID: 16996197.
- Ariel Dinar and Margaret Hogarth (2015), "Game Theory and Water Resources: Critical Review of its Contributions, Progress and Remaining Challenges", Foundations and Trends® in Microeconomics: Vol. 11: No. 1–2, pp 1-139. http://dx.doi.org/10.1561/0700000066

Thank you, both are relevant citations and we have included them in the revision.

https://hess.copernicus.org/preprints/hess-2021-312/#discussion

**Comments from Reviewer 2**
https://doi.org/10.5194/hess-2021-312-RC2

General comments:

The article combines game theory with groundwater modelling to analyse social dilemmas and potential policy solutions related with groundwater contamination. This is a timely contribution given that groundwater contamination remains a widespread challenge. The authors' detailed description of the  the game theoretical model and the groundwater modelling is informative and helpful - especially for readers not familiar with the framework proposed. I also find the application of the framework to a real-world scenario interesting. I believe further work on the case study and the discussion section would contribute to strengthen the paper. To this end, I provide some suggestions below.

Thank you for the careful consideration of our manuscript and for the constructive suggestions, which we believe will improve key aspects of the manuscript.

The authors show the relevance of the methodology presented by applying it to the case of St. Joseph County. I would suggest the authors to further contextualize the case study. For instance, it would be of interest to include more data about the socio-economic status of residents of the two sites considered (Centre Township and Granger CDP) and housing density as well as some more information about environmental policies in place to tackle groundwater use/contamination. This would be helpful to assess the potential of the framework to understand real-world cases. Recent work on water security in the context of the US has highlighted significant inequalities that I believe it is worth discussing either in the presentation of the case study or in the discussion session, see for instance:

- Meehan, K., Jurjevich, J.R., Chun, N.M. and Sherrill, J., 2020. Geographies of insecure water access and the housing–water nexus in US cities. Proceedings of the National Academy of Sciences, 117(46), pp.28700-28707.
- Meehan, K., Jepson, W., Harris, L.M., Wutich, A., Beresford, M., Fencl, A., London, J., Pierce, G., Radonic, L., Wells, C. and Wilson, N.J., 2020. Exposing the myths of household water insecurity in the global north: A critical review. Wiley Interdisciplinary Reviews: Water, 7(6), p.e1486.

The suggested references are highly pertinent and we now include both of them to provide additional perspective, in particular by shedding light on our additional discussion of policy as it pertains to St. Joseph County. As noted in the manuscript, both Centre Township and Granger Census Designated Place (CDP) exhibit high prevalence of nitrate contamination at the 5 ppm and 10 ppm levels. We will now also highlight additional similarities and differences between the two in the manuscript in the discussion.

We also note (L409):

As both regions are located within unincorporated areas of St. Joseph County, they face the same regulations with respect to housing density and well placement.

The discussion is limited to describe different social dilemmas and propose (rather vague) policy solutions. The author could work to further develop the discussion section with specific reference to the case study. For instance, I am missing a clear statement indicating to which extend the framework proposed is helpful to i) describe real-world social dilemma, ii) identify (applicable) policy solutions.

Thank you for this important comment, which was also reflected in the comments of the other reviewer. We now structure the discussion to (a) describe theoretical interpretations of the game (as before), (b) include policy considerations for local governments such as St. Joseph County (new text below), and (c) discuss limitations of the model and applications to other regions (new text in response to subsequent comment). We will include the following text in the revised manuscript to address this comment (L397):

Subsection: Policy considerations

The model identifies how social dilemmas arise through tension between the cost of well contamination and inability of individual households to prevent contamination. In particular, wealthier households may be able to collectively organize to prevent contamination through enhanced septic treatment. This approach can be facilitated by homeowners associations, and public education about the consequences of contamination. As the model demonstrates, the drawback of this framing is that lower-income households may be averse to participation in such projects because the costs of enhanced septic treatment exceed the economic benefits associated with home prices. This creates an obvious conundrum for local governments whereby the most equitable health outcomes cannot be achieved through community collective action because lower-income households are reluctant to participate in collective initiatives for enhanced septic treatment. It is further problematic in the U.S. where low-income households are less likely to be supplied by piped water (Meehan, 2020a) and more face greater environmental health risks (Meehan, 2020b) than high-income households.

We can consider whether or not this possibility may be occurring in St. Joseph county by comparing Granger CDP and Centre Township. As both regions are located within unincorporated areas of St. Joseph County, they face the same regulations with respect to housing density and well placement. We obtained income distributions for each location from http:\censusreporter.org, which aggregates census statistics by geographical areas and provides median income and four income groups ($0-50k,$50k-100k, $100k-200k, >$200k). Centre Twp has a median household income of $65k and the most common income bracket is $0-50k (39% of households) (Census Reporter, 2022a), whereas Granger CDP has a median income of $102k and the most common income group is $100k-200k (35% of households) (Census Reporter, 2022b). In Centre Twp, nearly all dense neighborhoods with over two houses per acre have at least 50% prevalence of contamination over 5 ppm (Fig. 7a-i). Conversely most dense neighborhoods in Granger with over two houses per acre have less than 50% prevalence of nitrate contamination over 5 ppm (Fig. 7a-iii). Thus, the relative prevalence of contamination in these two locations is consistent with the national trend where lower-income households face greater risks.

The above analysis suggests three potential policy approaches depending on the disposition of local residents. First, promoting homeowners associations that require

enhanced septic treatment would be acceptable to wealthier households provided this initiative is combined with sufficient public education. Lower-income households may oppose such requirements because the economic burden of installing enhanced treatment likely exceeds the perceived benefits. Second, local governments can require wastewater treatment, either via enhanced septic systems or public sewerage. This option will likely lead to more equitable public health outcomes but potentially inequitable economic outcomes that disadvantage lower-income households who rent or own property with lower values. Third, the local government can leverage a combination of taxes and fees to incentivize (i.e., subsidize) wastewater treatment via enhanced septic systems or community sewerage. The game theoretic model provides a first estimate of the subsidies that would be required such that the cost of treatment would match the reduction in property value from nitrate contamination. Public outreach could be used to refine the taxation structure and value of subsidies so that residents are amenable to this approach.

I would also suggest to further reflect on the limitations of the proposed framework especially in relation to policy solutions in a real-world scenario and to the uncertainties in the modelling. For instance, the authors assume the players are homogeneously located, do not have perfect information and they face similar costs and emit comparable pollution (l.356). How does such assumptions influence the results, especially in terms of identifying policy solutions? Loss in home value is considered as an estimate for the cost of contamination, while in the conclusion the authors suggest that more data are needed, they could also explain how does this assumption influence the findings. Moreover, I was wondering if and how the choice to focus on a case study located in the US (and not for instance a case in the Global South where groundwater contamination is also a pressing challenge) would influence the author's assessment of the potential of the framework to understand social dilemmas and advice policy making.

We agree that additional context would be beneficial. We will add text to the manuscript that incorporates the following points to address how the assumptions of our modeling approach relate to policy implications in St. Joseph County (and similar regions) in addition briefly describing how the model could be applied to scenarios that would be more likely to arise in low- and middle-income countries. The revised text reads (L430):

Subsection: Model assumptions and broader applications

This approach must take into consideration the assumptions and context in which the model was applied in St. Joseph County, including that (a) houses were uniformly distributed on a grid, (b) contamination occurred due to nitrate pollution, and (c) the cost of contamination was determined as the associated reduction in home prices.

The assumption of homogeneity yields situations of symmetry where the payouts are identical for all players. There are two features of heterogeneity that could affect model outcomes and therefore policy considerations. First, heterogeneity in the hydrogeology or housing density would create situations where some households are located upstream of the majority of the neighborhood and therefore would be less interested in contributing to collective action schemes that required enhanced septic treatment, as these households would be unlikely to be contaminated regardless of the behavior of other households.

However, these theoretical limitations are unlikely to affect the policy process for the following reasons. To our knowledge, the majority of situations with high community nitrate contamination occur in hydrogeological settings with unconsolidated aquifers and flat topography, meaning that the direction of groundwater flow may not be obvious. It would therefore place local governments in a tenuous position to attempt to adequately assess the groundwater flow paths in order to customize policy to such asymmetries, and although they may exist they can be effectively ignored in most situations. Second, heterogeneity in property values would create a dynamic environment where wealthier households are willing to pay more than less wealthy households for improved water quality. This should be taken into account when implementing policy, and may facilitate implementation of the third policy approach, described above.

The cost of contamination could just as well arise from any range of household contaminants. We focused on nitrate contamination, which is prevalent in St. Joseph County and many other locations in the United States, but other situations may arise with other types of pollutants and treatment strategies. The game can just as well be applied to these scenarios, accounting for different costs of contamination and treatment. For instance, pathogenic contamination is more prevalent in low- and middle-income countries, and the associated economic consequences include loss of work and income (Prüss-Üstün, 2016). Similarly, treatment strategies are likely to differ, both in terms of domestic water treatment and waste treatment. A complete assessment is beyond the scope of this manuscript, but the game theoretic model could readily be applied to such situations using the R package developed for this manuscript (Penny, 2021).

Prüss-Üstün, A., Wolf, J., Corvalán, C., Bos, R., & Neira, M. (2016). Preventing disease through healthy environments: a global assessment of the burden of disease from environmental risks. World Health Organization.

Minor comments:

Title: the authors might want to consider to specify that the study focuses on a case study located in the US.

We have made a minor change in the title, switching "private water systems" to "household water systems" in response to the comment below. We decided to keep the remainder of the current title because the theoretical aspects of the manuscript, including game theory and R package, apply generally to water quality in household water systems. We now briefly describe (in the discussion) how the approach could be applied to scenarios with other pollutants and economic considerations (as might arise in low- and middle-income countries).

l.9 – Repetition of 'three'

Thank you, we have fixed this.

l.25 – Here and in other sections the authors refer to "private water systems". Perhaps the term household water systems or domestic water systems would be more appropriate in order to distinguish the water systems considered in the paper and avoid confusion with i.e. a privatized pipe-born networked systems.

We agree with this suggestion and have revised the paper to refer to household water systems rather than private water systems.

I. 320-321 – It is unclear to what the letters B and E refer to.

We have clarified that B refers to a "basic" (i.e., traditional) septic system and E refers to enhanced septic treatment that would remove contaminants of concern. For instance, advanced septic systems (e.g., the Aquapoint, Inc., Bioclere™ 16/12 system, noted in the case study) can remove nitrate contamination from septic leachate.